# Transmural Ultrasonography in the Evaluation of Horse Hoof Internal Structures: Comparative Qualitative Findings—Part 1

**DOI:** 10.3390/ani13121951

**Published:** 2023-06-10

**Authors:** Andrés Felipe Castro-Mesa, Rafael Resende Faleiros, José Ramón Martínez-Aranzales

**Affiliations:** 1Equine Medicine and Surgery Research Line (LIMCE), CENTAURO Research Group, School of Veterinary Medicine, Faculty of Agricultural Sciences, Universidad de Antioquia, Medellín 050010, Antioquia, Colombia; jose.martinez@udea.edu.co; 2Equinova Research Group, School of Veterinary, Universidad Federal de Minas Gerais, Belo Horizonte 31270-901, Brazil; faleirosufmg@gmail.com

**Keywords:** ultrasonography, transmural technique, hoof dermis, lamellar layer, sublamellar dermis, third phalanx

## Abstract

**Simple Summary:**

Various imaging techniques are used to assess the structures of equine limbs and hooves. However, each technique has its limitations. Ultrasonography, for instance, is not currently considered sufficient for fully evaluating the dermoepidermal junction (DEJ) of the hoof. To address this limitation, we hypothesized that immersing the hoof wall in water could produce better results. After a 24 h period of immersion of the digit, the normal appearance of the middle stratum (tubular wall), the internal stratum (lamellae), the sublamellar dermis, and a few structures of the distal phalanx could be described using a transmural technique. Our findings demonstrate that a transmural technique, used after a 24 h immersion of the digit, permits the ultrasonographic identification and characterization of the middle stratum (tubular wall), internal stratum (lamellae), sublamellar dermis, and some structures of the distal phalanx. This transmural technique may have specific utility in evaluating structures involved in hoof conditions, particularly laminitis.

**Abstract:**

Ultrasonography is commonly used to evaluate equine lameness-related structures, but traditional hoof approaches cannot adequately visualize internal structures such as the lamellar layer. To address this limitation, we used 62 healthy hooves without digital radiographic abnormalities, made up of thirty anatomical pieces (phase 1) and thirty-two hooves from living horses (phase 2). In both phases, half of the digits were submerged in water (group 1) and the other half in water and ice (group 2) for 24 h. Ultrasonographic views and temperature and humidity measurements were taken every two hours, and sagittal sections were obtained in phase 1. Good quality images were obtained in both phases. The transmural technique allowed the evaluation of several structures of the DEJ (tubular and lamellar strata of the hoof wall, sublamellar dermis, and solar and coronary papillae) and of the distal phalanx (extensor process, parietal surface, and apex). Our findings suggest that the transmural technique following hoof submersion can be useful for internal evaluation of hoof conditions, such as laminitis. However, further validation of the technique in natural or experimental cases is required.

## 1. Introduction

Ultrasonography is a diagnostic technique widely used to evaluate structures associated with lameness in the distal part of the limbs of horses [1]. Several approaches to assessment of structures within the hoof have been described [2,3,4,5,6,7,8,9,10,11,12]. However, none fully evaluates the dermis, the epidermis, and the dermoepidermal junction (DEJ) of the digit. Furthermore, obtaining diagnostic images through the keratinized tissue of the hoof has been challenging in previous studies [6,7,10], and as a result, an ultrasonographic window through the wall has not been described before.

The parietal DEJ—more commonly known as the internal (lamellar) layer of the wall—is the interdigitation between the dermal and epidermal layers of the hoof, which makes up the suspensory apparatus of the distal phalanx (SADP), which aims to stabilize the digit and protect it from load forces [13,14,15]. As a result, changes to the lamellar layer in cases of laminitis cause the collapse of the distal phalanx in the majority of cases [16,17,18,19,20,21]. Consequently, the lamellar layer is a critical site of interest for early assessment during clinical approaches. Nevertheless, routine diagnostic methods have significant limitations in detecting changes in soft tissues [22,23,24], necessitating the use of advanced imaging techniques such as magnetic resonance imaging (MRI) and computed tomography (CT) [13,24,25]. However, these techniques are still of limited use due to problems with equipment availability and high costs [22,26].

The diagnostic value of images obtained of the lamellar layer and the sublamellar dermis requires the recognition and characterization of their normal status, to interpret changes derived from clinical conditions such as laminitis. The parietal dermis is divided into two layers: (1) the lamellar dermis—composed of primary and secondary dermal lamellae that are attached to the basement membrane—and (2) the sublamellar dermis—composed of sublamellar blood vessels and collagen fibers that connect to the foramina on the distal phalanx parietal surface, suspending it inside the hoof [13,14,27]. This information is relevant in laminitis cases, when the detection of changes in the developmental phase becomes more relevant [28,29] and limited evidence is available in this regard [22,23].

Based on our pilot studies, the objective of this research is to describe a transmural ultrasonographic technique that can evaluate the internal structures of the equine hoof after a submersion period. We also aim to determine the optimal immersion time in the conduction medium and the positioning of the transducer to obtain high-quality diagnostic images.

## 2. Materials and Methods

### 2.1. Anatomical Pieces and Hooves from Living Horses

This research was approved by the Ethics Committee for Animal Experimentation of the Universidad de Antioquia, Colombia (Act number 135, 2 September 2020). Sixty-two healthy hooves from thoracic limbs (TL) and pelvic limbs (PL) of Colombian creole horses were divided in two study phases. In phase 1, 30 hooves collected in a slaughterhouse were disarticulated from the fetlock joint, transported fresh for 4 h, and finally frozen at −21 °C for 150 days. In phase 2, 32 hooves from eight healthy mares without lameness and with an average age, weight, and body condition score [30] of 6.9 ± 2.97 years, 284.9 ± 32.1 kg, and 4.1 ± 0.44, respectively, were selected based on their normal condition at clinic inspection, including criteria for body condition score [30]. Only clinically healthy hooves were used, based on a negative lameness inspection that involved assessing the lameness at walk and light trot, and using hoof testers and percussion. These inspections were carried out on all horses, including horses that were the donors of the pieces. All hooves were also evaluated using digital radiography to rule out any alterations in internal and external structures. Lateral–medial projections were taken using a portable veterinary X-ray equipment, generator MINXRAY^®^ HFX90V (MinXray, Inc., Northbrook, IL, USA), with 1.2 mAs–66 kv settings. The location and normal anatomy of the distal phalanx were determined by measuring angles and other factors (radiological software AccuVet^®^ version 1.1.0 developed by RadmediX) with relevance to laminitis [29,31,32,33,34,35,36].

In phase 1, a total of 16 hooves (4 TL and 4 PL by group) were randomized, selected and sectioned in a sagittal plane using a band saw (JAVAR^®^ v25-19, 220v/60/2f-1hp). The sagittal sections were measured and analyzed with a digital stereomicroscope (Olympus^®^ SZX7, DP27 camera, manufactured by Evident Corporation, Inatomi, Tatsuno-machi, Kamiina-gun, Nagano, Japan; and Olympus^®^ cellSens Standard software version 2.3), as shown in Figure 1. These anatomical sections were used as a reference for the evaluation and interpretation of the ultrasonographic findings to differentiate between the lamellar layer (stratum internum) and the sublamellar dermis. These are two separate components of the parietal dermis [13,14,17,35,36], which cannot be distinguished by digital radiography [33].

In addition, the normal configuration of the dermal and epidermal layers of the hoof was evaluated using a contrast-adjusted technique, which allowed for the detection of both radiolucent and radiodense appearances [33,34,36].

Two immersion mediums were utilized in both phases of the study: water in group 1 (G1) and a mixture of ice and water in group 2 (G2) (Figure 2). These mediums also served as conduction mediums during the ultrasonographic evaluations. The number of hooves for each group was as follows: for phase 1, G1 had 8 TL and 7 PL while G2 had 7 TL and 8 PL; for phase 2, each group had 8 TL and 8 PL, which translates to 4 mares per group.

Before being immersed for 24 h, each hoof was thoroughly cleaned and sanded. The bags containing the commercial solutions (with a capacity of 3 L) were changed as needed to ensure that the immersion process was adequate. Throughout the immersion period (every 2 h), ultrasonographic evaluations were carried out using a G30 Color Doppler ultrasound machine (EMP^®^ Shenzhen Emperor Electronic Technology Co., Ltd. Nanshan district, Shenzhen, China) and the temperature measured with an infrared thermometer (UX-A-01, Big Healthy^®^, China) and the hoof surface humidity measured with a hygrometer (MT270, Tavool^®^) were recorded.

### 2.2. Transmural Ultrasonographic Technique

The ultrasonographic examinations were always performed by the same researcher, who was aware of the experimental group. The exam was performed using a 6.5 MHz linear transducer for tendons, which was submerged in water or ice and water, depending on the assigned medium, without a standoff pad. The transducer was positioned in the sagittal plane of the hoof wall to obtain images of the distal phalanx (Figure 3a). The identification of the distal interphalangeal joint (DIPJ) and the extensor process were accomplished before exploring distally, determining the change in inclination between the extensor process and the parietal surface as the proximal level [33]. The distal level was located approximately 6 mm proximal to the apex of the distal phalanx [36]. Using the transverse view on the coronary band (Figure 3b) [9], the spatial relationship between it and the extensor process was established.

### 2.3. Image Evaluation and Classification

The evaluations of the images obtained through digital radiology, ultrasound, and digital stereomicroscopy were conducted by a single researcher during the examination procedures. The researcher assessed the ability to differentiate the middle layer (tubular), the inner layer (lamellar), and the sublamellar dermis of the hoof, along with the transitions between them, at the proximal and distal levels of the third phalanx, to categorize the images. An image was deemed of high quality only when all these structures could be clearly differentiated.

### 2.4. Statistical Analysis

Descriptive statistics were first calculated as measures of central tendency (i.e., interquartile range (IQR), mean (average), median) and measures of dispersion (i.e., standard deviation (SD), coefficient of variation (CV)). For non-normally distributed data, the analyses were performed based on the median (50th percentile). Using Pearson’s chi-square (χ^2^) nonparametric test, the association between temperature, humidity, and immersion time (ordinal quantitative variables) and the prevalence of high-quality ultrasound images (nominal categorical variable) was estimated, considering a significance level of *p* ≤ 0.05. Data were analyzed using STATA^®^, version 16.1 (StataCorp., College Station, TX, USA).

## 3. Results

The number of high-quality images obtained with digital radiography, ultrasonography, and stereomicroscopy of sagittal sections in both phases is shown in Table 1. In phase 1, the relative prevalence of high-quality ultrasonographic images was 73% in G1 and 20% in G2, while in phase 2, it was 50% in G1 and 69% in G2. As for the anatomical sections, G1 and G2 presented 20% and 27% high-quality images, respectively. Finally, all radiographic images were of high quality.

Table 2 shows the variables considered for the description of the ultrasonographic technique (i.e., values for immersion time, hoof humidity, and hoof temperature), aimed at obtaining high-quality images. According to the CV, humidity percentage was the most homogeneous variable in both phases, and the most heterogeneous was temperature. In general, data from phase 2 were more heterogeneous.

Association test results for each variable that could influence the transmural technique’s ability to obtain high-quality images are shown in Table 3. The analyses determined a statistically significant association (*p* ≤ 0.05) with the immersion time of G1 in phase 2, an approximate time of 7 h. Although the immersion time of G2 was similar, no statistical significance was found.

Regardless of the position of the transducer, the experimental phase, and the conduction medium used, the transition between the middle layer (tubular), the inner layer (lamellar), and the sublamellar dermis could be identified at the proximal and distal levels of the distal phalanx after the preparation procedures. At the proximal level and close to the extensor process, the coronary papillae were observed to be parallel to the parietal surface, indicating the normal anatomical configuration of the hoof-wall growth (Figure 4c). However, in some cases, these structures were inclined (Figure 4b) due to incomplete contact between the surface of the coronary band and the transducer.

At the distal level, it was found that the sole papillae were located distal to the sole border (Figure 5b). Such findings were represented by interrupted oblique lines, hyperechoic to the sole dermis, and then later, isoechoic with some blood vessels. The reliability of being able to determine this level increases when observing the sole papillae.

Two transition zones can be seen, in both Figure 4 and Figure 5. The first is located between the middle layer (tubular wall) and the inner layer (lamellar) of the epidermis—differentiating it, with greater echogenicity than that of the previous ones. The second is located between the lamellar dermis and the sublamellar dermis. The appearance of this transition zone is reinforced by the contact of ultrasound waves with the parietal surface of the distal phalanx, and varies at the distal level at the end of the phalanx, with echogenicity being less than that of the anterior section (Figure 5b); the change is also observed in the sublamellar dermis.

Once the structures inside the hoof were located, their textures were described (Figure 4 and Figure 5). The tubular hoof wall shows a heterogeneous and anechogenic appearance adjacent to the transition to the inner layer (lamellar). The lamellar layer has an anechoic appearance (occasionally with fine granules). The sublamellar dermis has a heterogeneous appearance, being composed of blood vessels (arterioles/venules); however, it was not possible to determine the blood flow in the phase 2 hooves using Doppler. The sublamellar dermis is located between the DEJ and the periosteum (fibrocartilage) of the parietal surface of the distal phalanx [14]. The parietal dermis is composed of the previous two.

## 4. Discussion

The ultrasonographic technique through the hoof wall (transmural technique), as described and implemented in this study, allowed the identification and delimitation of tissues inside the hooves of normal limbs, including structures of interest such as the lamellar layer (which contains the DEJ), previously observed in studies using MRI and CT [23,37,38]. This raises expectations for the use of transmural ultrasonography in cases of clinical conditions that compromise this structure, such as laminitis. It is an easily accessible and repeatable technique that can aid follow-up research. Meanwhile, it is important to note that the immersion in water and increased humidity of the keratinized tissue is essential to overcome the limitations described in previous studies [6,7,10,12]. The immersion of the hoof in water or water and ice sought to rapidly hydrate the keratinized tissue, altering the polymer chains of the keratin matrix and causing greater freedom and less rigidity [39], which would allow access to the ultrasound waves. The benefit of hydrating the hoof has been used for sole and frog approaches in previous studies, which have evaluated internal structures different from those reported herein [1,3,4,5,6,8,11]. However, in addition to the immersion medium, the exposure time is essential for obtaining images of diagnostic quality.

Although high-quality images were obtained using both immersion mediums, an important difference was observed regarding the immersion time, in that phase 2 hooves required at least 7 h, compared to 16 h in phase 1. Phase 2 hooves kept the hydration mechanisms intact and active [39], allowing a better regulation of the humidity degree. This is corroborated by the higher percentage of hooves that produced high-quality images (69%) and the significant association between image quality and immersion time demonstrated in this phase. The greater time in phase 1 was possibly influenced by the freezing state of the anatomical pieces, since freezing increases tissue density and the crystallization of fluids, limiting the entry of the ultrasound waves. However, this time was less than the 18–24 h thawing period previously seen in studies using MRI [23,36].

Despite the association between different immersion times and obtaining better images with ultrasonography, humidity was the most homogeneous variable while temperature was the most heterogeneous in both phases. When ice was used in phase 2, the variability in temperature tended to be higher, possibly due to the activation of the digit thermoregulation mechanisms [40,41]. The normal function of the arteriovenous anastomoses (AVAs) is to maintain the temperature of the hoof since a double circulation is occurring (i.e., a slow one that provides nutrients to maintain the metabolism of the lamellar tissue, and a fast arteriovenous anastomose (AVA) circulation that periodically supplies warm arterial blood to the hoof dermis when it reaches low temperatures) [41]. These mechanisms could have increased the humidity degree and the temperature of the hoof dermis, possibly contributing to image quality (Table 1). The study did not allow this to be fully elucidated (Table 3), since the temperature was measured on the wall surface and the temperature of the immersion medium was not controlled. For future studies, and with the intention of obtaining better associations, it is suggested that the variation of the hoof temperature and humidity be estimated. However, high-quality images were obtained independent of the phase and the immersion medium, indicating that both mediums were effective for the technique.

Although maintaining constant temperature control can be challenging, one of the reasons for using water and ice as a conduction medium was its practicality since digital cryotherapy is recommended and widely used in cases of acute laminitis [42,43,44,45,46]. The 24 h immersion time of the hooves of living horses in water and ice did not cause any adverse effects, as reported by previous studies of 48 h of cryotherapy [47]. Apparently, complications would appear after 72 h of immersion and depend on the method [46,48].

The benefit of using anatomical pieces was the ability to confirm and validate the strata/layers obtained by the radiographic and ultrasonographic studies through anatomical sections. However, the use of frozen pieces made it difficult and delayed the obtaining of ultrasonographic images, as they required a thawing time that was delayed when a water and ice mixture was used as a conduction medium. It is possible to obtain better effects with fresh pieces, although the animal study showed better results.

Despite some limitations in methodology, such as the researcher performing all the examinations and image evaluations being aware of the sample group, the present study successfully described and characterized the acquisition of ultrasonographic images of the middle stratum (tubular wall), internal stratum (lamellar), and sublamellar dermis in healthy hooves. Its findings encourage the application of this technique in cases of natural or induced laminitis for further validation.

In the evaluation of the images obtained using the transmural technique at the proximal level, the coronary papillae were clearly visible. These structures have also been observed in MRI and CT imaging studies [37,38] and reported using contrast-adjusted digital radiography [34]. Configuration changes in these and in the tubules’ growth direction in the coronary region are relevant findings in the sinking of the distal phalanx in the chronic phase of laminitis [49] that could be observed, in addition to MRI, using the transmural ultrasonographic technique described herein.

The concave shape of the coronary band made it difficult to complete the contact with the transducer, producing changes in the inclination of the coronary papillae in the image in Figure 4b. However, when descending the wall, their normal positioning, parallel to the parietal surface of the distal phalanx, was observed (Figure 4c). This obstacle was also previously described in studies using ultrasonography to assess the collateral ligaments of DIPJ [10]. The use of a standoff pad may correct the lack of contact at the proximal level between the transducer and the coronary band. Continuing with the proximal evaluation, the periople (external layer of the epidermis) was only observed adjacent to the coronary band. This is due to progressive wear as it descends towards the distal part of the hoof [27,39,50].

At the distal level, the images revealed the inclination of the dermal papillae, which plays a crucial role in determining the growth pattern of the sole epidermis. Similar to the coronary papillae, the horn tubules also join the dermal papillae [51], which explains the interrupted anechoic appearance observed in Figure 5. Additionally, the slope of the dermal papillae is parallel to the transition between the sublamellar dermis and the lamellar dermis, which provides unprecedented structural details. On the other hand, the refraction phenomenon could limit the appreciation of the circumflex artery or venule. When the ultrasound beam is not perpendicular to tissue boundaries, the anatomy may be dislocated and/or not reached [52].

The heterogeneous appearance of the middle layer (tubular wall) [51] (Figure 4 and Figure 5) is due to the tubules being organized by zones according to density [41,53]. That is, the outer layer is the most rigid (high density of tubules) and the inner layer is the most flexible (low density of tubules) [54]; in turn, and reflecting the water content, the inner layer is the most hydrated [39]. This explains the anechogenic appearance adjacent to the transition between the middle (tubular wall) and the inner (lamellar) layers [51] of the wall (Figure 5). The middle layer tubules have also been described using MRI and CT [37,38], but without differentiating layers.

The parietal dermis can be divided into two layers. The lamellar dermis is composed of primary and secondary dermal lamellae that are attached to the basement membrane, and the highly irrigated sublamellar dermis, composed of sublamellar blood vessels and collagen fibers [14,41,50,51] that connect to the foramina from the parietal surface of the distal phalanx, suspending it inside the hoof [13,14,27]. The parietal surface of the distal phalanx is provided with a modified periosteum [14] that presents a partially calcified fibrocartilage plate [50]; therefore, it does not reflect the characteristic (hyperechoic) appearance (Figure 5). Similar findings have been reported with MRI [23,37].

The transmural technique allowed the differentiation of the layers that make up the lamellar and sublamellar dermis [51], as well as the respective homogeneous and heterogeneous appearance. Such findings have been previously reported [23] in studies using MRI on hooves without alterations or considered normal. However, it was impossible to differentiate dermal and epidermal lamellae from the crisscrossing pattern that characterizes the lamellar layer, despite the vascular and corneous composition, respectively [13,55]. The image obtained was anechogenic in appearance. The blood vessels in the dermal lamellae are small, in the order of micrometers [40], which is why they were not visualized.

Ultrasonographic evaluation of the sublamellar dermis revealed the presence of larger caliber vessels (Figure 4 and Figure 5), also reported with MRI, CT, and digital venography [23,37,56,57,58]. However, Doppler with a linear transducer failed to reveal blood flow. Possibly, the ultrasound waves were incident at an angle of 90° (Doppler angle) on the parietal surface. In Doppler ultrasonography, the blood velocity (moving object) is equal to the current velocity multiplied by the cosine (cos) of the Doppler angle, the cos of 90° = zero (0); therefore, the flow is not detected [59]. The inclination of the transducer, and the Doppler angle, would have varied with a gel pad (standoff pad), producing a different result, and the blood vessels could have been visualized using a “phased array” transducer [60].

The technique implemented herein revealed images of structures that are usually affected in cases of laminitis [16,17,18,19,20,21]; therefore, it was possible to characterize and describe details of their normal ultrasonographic anatomy, the recognition of which could be important and applicable in early diagnosis and monitoring of clinical cases. This was corroborated by an incidental finding that occurred during the search for horses to be included in this assay. An ultrasonographic evaluation of a case of acute laminitis revealed a variation in the ultrasonographic appearance of the parietal dermis, limiting the detection of the respective transitions [61]. Likewise, MRI detected structural changes and changes in signal intensity of the lamellae and sublamellar portion in acute laminitis, without radiographic alterations, but with high histological correlation [23].

Discrepancies have been found between clinical and radiographic findings—basic diagnosis to assess severity and prognosis in laminitis, and null information on soft tissues [24]—demonstrating the significant limitations of these methods [22,23]. This led to the use of MRI and CT as the best options [24,25]—or positron emission tomography (PET)—because of their ability to differentiate acute and chronic cases of laminitis [62]. Nevertheless, limitations have been described in the use of these techniques, such as economic factors and problems with equipment availability, restricting their use [26,34].

MRI is superior to digital radiography for determining normality and changes in the hoof soft tissues [12,23,37]. In addition, radiographic projections only detail the dermis, without differentiating between the lamellar and sublamellar portions [33,36], something which ultrasound through the wall has been able to do. However, digital radiography was implemented in the present study to assess the spatial normality between the distal phalanx and the hoof and to determine the normal configuration of the dermis and the normal anatomy of the distal phalanx [29,31,32,33,34,35,36,63], since, as an inclusion criterion, the hooves had to be unaltered. On the other hand, transmural ultrasound imaging produced results similar to those described when using MRI through the hoof wall. The comparison between these two diagnostic aids has already been made with the evaluation of the deep digital flexor tendon (DDFT), finding greater sensitivity to ultrasound of lesions in the proximal recess of the navicular bursa and equal accuracy in detecting adhesions of the DDFT inside the hoof [64,65].

## 5. Conclusions

The transmural technique, used after a 24 h immersion of the digit in water, permitted the ultrasonographic identification and characterization of several structures of the DEJ (tubular and lamellar strata of the hoof wall, sublamellar dermis, and solar and coronary papillae) and of the distal phalanx (extensor process, parietal surface, and apex). In addition, a statistical association between high-quality images and submerging the hooves of living horses in water for seven (7 ± 1.4) hours was found. Likewise, quality images were obtained using water and ice as the immersion medium.

To the authors’ knowledge, this is the first time this technique has been described and used. To increase its reliability, it is necessary to contrast the findings of this study with magnetic resonance imaging (both are able to differentiate the components of the hoof dermis and epidermis). Although preliminary, our findings suggest that the transmural technique following hoof submersion can be useful for internal evaluation of hoof conditions, such as laminitis. However, further validation of the technique in natural or experimental cases is required.

## Figures and Tables

**Figure 1 animals-13-01951-f001:**
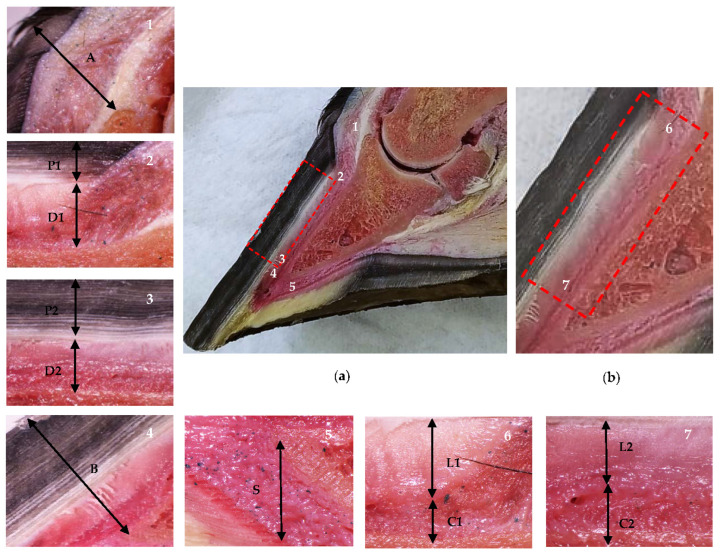
Anatomical sagittal plane section of an equine thoracic hoof. Proximal is to the right in images **2**–**7**. (**a**) Dermal and epidermal tissues of the hoof and the distal phalanx parietal region. In the detailed images, note the coronary band to extensor process distance (**1**, A), the tubular hoof wall (**2**–**3**, P1 and P2), and the lamellar layer and the sublamellar dermis (**2**–**3**, D1 and D2) at the proximal (**2**) and distal (**3**) levels, respectively. Note the distal phalanx apex to hoof wall distance (**4**, B) and the sole dermis (**5**, S). (**b**) In the enlarged detail, note the lamellar layer (**6**–**7**, L1 and L2) and the sublamellar dermis (**6**–**7**, C1 and C2) at the proximal (**6**) and distal levels (**7**), respectively.

**Figure 2 animals-13-01951-f002:**
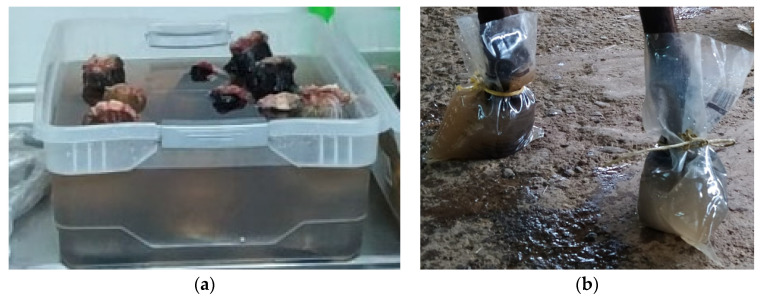
Hooves immersion techniques. (**a**) Plastic containers for frozen anatomical pieces. (**b**) Plastic bags of commercial solutions (3 L capacity) for living horses.

**Figure 3 animals-13-01951-f003:**
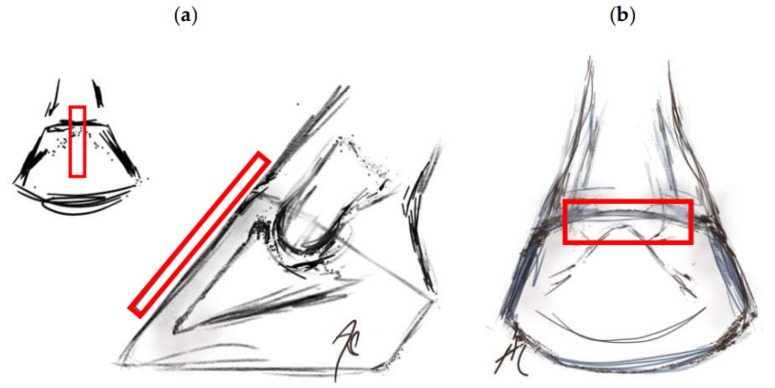
Location of the transducer in the transmural ultrasound technique (rectangle). (**a**) Transducer positioned in the sagittal plane, identifying the distal interphalangeal joint (DIPJ) and the extensor process. (**b**) Transverse section at the coronary band level.

**Figure 4 animals-13-01951-f004:**
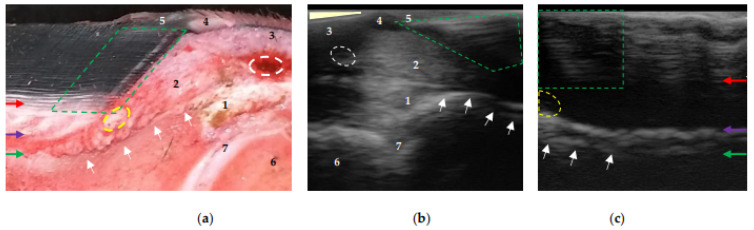
Equine hoof internal structures at the proximal level, observed using a transmural ultrasonographic technique on equine hooves, and their correspondence to the anatomical section. (**a**) Sagittal anatomical section of the wall of a thoracic limb which was filled through the median artery. Proximal is to the right. (**b**) Sagittal wall ultrasonography immediately distal to the distal interphalangeal joint (DIPJ) of a pelvic limb. Proximal is to the left. (**c**) Transmural (sagittal) wall ultrasonography of a pelvic limb. Proximal is to the left. The transition between the middle (tubular wall) and inner (lamellar) layers of the epidermis (red arrow). The transition between the lamellar dermis and the sublamellar dermis (purple arrow). Between these two arrows is the lamellar layer. Periosteum (fibrocartilage) of the distal phalanx (green arrow), which, at the level of the extensor process (white arrows), presents the characteristic hyperechoic appearance at the insertion of the common/long digital extensor tendon (1). Coronary papillae (broken green area). Coronary artery (broken white circle). Coronary vessels (interrupted yellow area). 1: common/long digital extensor tendon. 2: hypodermis (pad) of the crown. 3: skin. 4: perioplic dermis. 5: periople (external layer). 6: middle phalanx. 7: DIPJ. The area where the transducer loses contact (light yellow triangle) in (**b**).

**Figure 5 animals-13-01951-f005:**
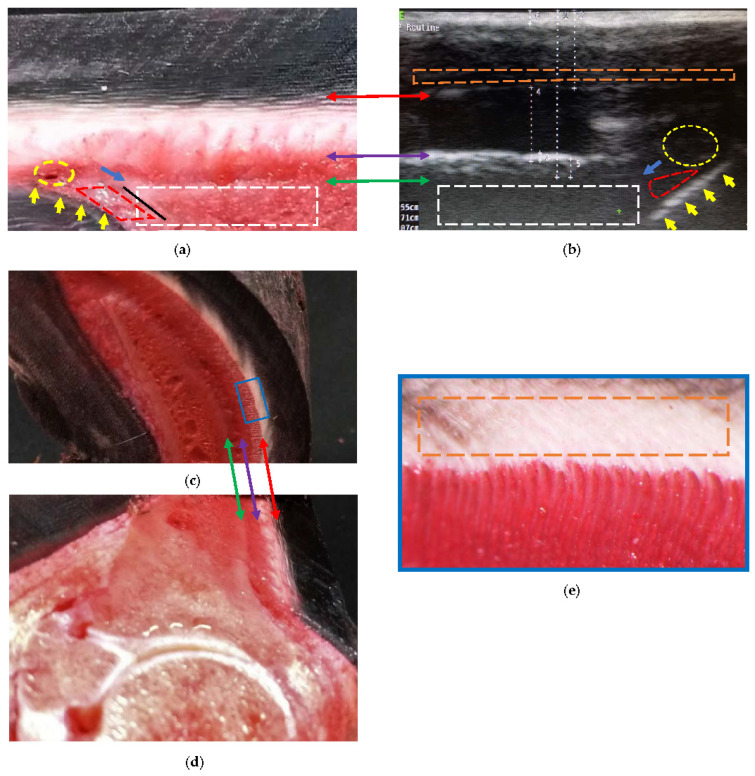
Comparative findings of the hoof internal structures at the distal level, observed using a transmural ultrasonographic technique on living equine hooves, and their correspondence to the anatomical section. (**a**) Sagittal anatomical section of the wall of a thoracic limb which was filled through the median artery. Proximal is to the right. (**b**) Ultrasonography in a thoracic limb of a living equine hoof. Proximal is to the left. The acoustic shadow produced by the bone mass (broken white box) of the parietal surface of the distal phalanx. The sole surface of the distal phalanx (black line) was not seen on ultrasound. Apex of the distal phalanx (blue arrow). Sole papillae (yellow arrows) that limit the sole dermis (broken red lines). Circumflex artery or venules (broken yellow circle). (**c**,**d**) Transposition between the sagittal and transverse anatomical sections. (**e**) Enlarged box, where the innermost region of the middle stratum of the epidermis can be seen (interrupted orange box). The transition between the middle (tubular wall) and inner (lamellar) layers (double red arrow) of the epidermis. The transition between the lamellar dermis and the sublamellar dermis (double purple arrow). Between these two arrows is the lamellar layer. Better detail of the sublamellar vessels and the lamellar layer is obtained in the transverse section. Periosteum (fibrocartilage) of the distal phalanx (double green arrow), with the respective acoustic shadow produced by the bone mass (broken white box).

**Table 1 animals-13-01951-t001:** Number of high-quality images of anatomical pieces (Phase 1) and hooves from living horses (Phase 2), according to the immersion medium, using digital radiography (DR), ultrasonography (US), and stereomicroscopy of sagittal sections (SS).

Group	Hoof	DR	US	SS
Phase 1 *				
G1	TL	8	5	0
	PL	7	6	3
G2	TL	7	0	1
	PL	8	3	3
Phase 2 **				
G1	TL	8	5	-
	PL	8	3	-
G2	TL	8	6	-
	PL	8	5	-

* Studies on anatomical pieces (*n* = 30 hooves). ** Studies on living horses (*n* = 32 hooves). G1: immersion in water. G2: immersion in a mixture of ice and water. TL: thoracic limbs. PL: pelvic limbs.

**Table 2 animals-13-01951-t002:** Values for the immersion time (hours), surface humidity (%) and temperature (°C) of the horse hooves subjected to transmural ultrasound.

Group	Hoof	Immersion Time	Humidity	Temperature
		Median ± SD	Median ± SD	Median ± SD
Phase 1 *				
G1	TL	16 ± 1.8	25.6 ± 2.4	16.1 ± 4.7
	PL	16 ± 1.8	25.6 ± 2.8	15.7 ± 4.7
G2	TL	16 ± 2.0	26.3 ± 2.8	9.2 ± 4.5
	PL	16 ± 1.8	26.3 ± 2.5	8.9 ± 4.6
Phase 2 **				
G1	TL	7 ± 1.4	26.7 ± 5.8	28.4 ± 8.9
	PL	7 ± 1.6	25.7 ± 5.2	28.4 ± 8.5
G2	TL	7 ± 1.5	23.2 ± 6.2	24.2 ± 10.6
	PL	7 ± 1.6	24.6 ± 6.8	24.2 ± 10.8

* Studies on anatomical pieces (*n* = 30 hooves). ** Studies on living horses (*n* = 32 hooves). SD: Standard Deviation. G1: immersion in water. G2: immersion in a mixture of ice and water. TL: thoracic limbs. PL: pelvic limbs.

**Table 3 animals-13-01951-t003:** Evaluation of the associations between humidity (H), temperature (T), and immersion time (IT) and the obtaining of high-quality images (HQ) in the transmural ultrasound.

Associations	Group	Degrees of Freedom	Statistic Test χ^2^	*p* Value *
Phase 1 *				
HQ–H	G1	11	11.55	0.40
	G2	11	15	0.18
HQ–T	G1	10	14	0.17
	G2	11	10.83	0.46
HQ–IT	G1	6	10.91	0.09
	G2	5	5.83	0.32
Phase 2 **				
HQ–H	G1	13	12.99	0.45
	G2	15	16	0.38
HQ–T	G1	12	10.98	0.53
	G2	14	13.67	0.47
HQ–IT	G1	5	12.32	**0.03**
	G2	5	3.53	0.62

* Studies on anatomical pieces (*n* = 30 hooves). ** Studies on living horses (*n* = 32 hooves). Pearson’s χ^2^ test, level of significance: *p* ≤ 0.05 *. Bold *p*-values are statistically significant. Group 1: immersion in water. Group 2: immersion in a mixture of ice and water.

## Data Availability

Additional data can be found from the publicly accessible repository of the Universidad de Antioquia at https://hdl.handle.net/10495/32382, accessed on 1 April 2023.

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
