# Peer review of "Transmural Ultrasonography in the Evaluation of Horse Hoof Internal Structures: Comparative Qualitative Findings—Part 1"

_animals, 2023, doi:10.3390/ani13121951_

Round 1

Reviewer 1 Report

Despite the good intentions of the authors to improve the diagnostic imaging of the equine hooves especially when laminitis is a possible com0lications, the paper needs some major revisions. Besides extensive English language editing, I would like to highlight some suggestions:

line 25-26: how Many for each group?

line 28: how many anatomical pieces an live horses?

the aim of the study must be better clarified. It is too general

line 84: this doesn’t rule out other underlying disorders 

line 80: how do you know they come from horses free from hoof disorders?

line 89: which associated tissues?

line 90 angle of what? Be more precise please

line 103: check for characters

fig 1 legend: it is not clear form who itch image the details belong to

Fig 2 add ()

line 112: i think you should better  describe how yoou manage to keep immersed in water the hooves for 24 h.. plastic bags are not so resistant

line 118: the probe submerged in what?

fig 3 ()

line 138: not normally distributed

line 140: you have to specify how the quality of US was categorized and according to what

line 166: if this is a trend and not significant, it is not of the same  importance as the previous one

line 180-182-183 (Figure….)

fig 5 legend: despite the excellent quality of the images, you need to be more detailed in the description of the images, in particular box B.

line 118-226-270-271-31 3 (figure..)

Line 274: a study must be reproducible. And I think that the lack of control of temperature (which is one of your variable, as said in M and M) is one of the major limitation of the study.

Reviewer 2 Report

An interesting and well presented study. There are several sentences which would benefit from editing. If the authors can contact a teacher of English, these small edits can be easily made and will improve the paper. 

On line 241, I would suggest changing 'corneal tissue' to 'keratinised tissue'.

Reviewer 3 Report

This is a very interesting paper concerning the use of Ultrasonography in the evaluation of Equine Hoof. The importance of the study is high, has an extensive anatomical description, and bibliography is sufficient.

Tittle  

Horse could be more appropriate than equine

Evaluation could be more appropriate than Definition

Simple summary

Intro and justification could be shorter and more focused. 

please describe very briefly the methodology

Could you try to formulate a hypothesis?

Why did the authors use different techniques for Hooves immersion? I mean why did the authors not use the plastic bags in the anatomical pieces?

94 “The findings of the anatomical cuts and radiographs were compared with those obtained by ultrasonography.”

It is not clear what comparison the authors made  of the anatomical cuts and radiographs. This is not expressed in the statistical part. please define findings, and if the name is to be a variable, avoid using it in any other way

L120 2.3. Transmural ultrasonographic technique

Here please describe the technique. the organization of the groups, although it is related to the preparation, should not fall under this heading. Or you could change the heading

The experimental design is not clear. What they did is clear, but I don't quite understand why they did it. For example, the cuts were made for what?

the discussion is nice. the limitations of practical application were well exposed. The anatomical atlas is impressive, and the anatomical, histological and ultrasound description is very well done.

“ finding a statistical relation with the quality images after submerging live horse hooves in water;” in the conclusions should be more careful. This is not supported by the results. immersion time is not the same as immersion.

Round 2

Reviewer 1 Report

The authors aimed to assess the feasibility of transmural US of the hoof wall to visualize dermis epidermis and epidermal junction. feasibility of the technique could be valuable as an adjunctive diagnostic aid for foot disorders such as laminitis.

I think that the topic is relevant as US is a relatively cheap technique, differently from MRI, and easy to be accomplished by most vets. 

Tables are well explained. Figures have a very high quality and their legend are well explained.

Overall methodology is good but I would like the authors to address a deficiency in methods. was the evaluation of image quality carried out by a single observer ? was the evaluation blind to the "treatment group"? was it carried out as soon as the exam was finished or after a while? I think that in a feasibility study repeatability and reproducibility are important to be evaluated, especially in US, where the exam is operator dependent. I would ask the authors to address these aspects in the methods section or in the discussion. 

  •  

Reviewer 3 Report

The reviewer would like to thank the authors for providing additional analysis and discussion of their research data and for making extensive additions to the manuscript to address comments from the first round review. The authors have addressed all recommendations for revision, and therefore the reviewer recommends accepting the manuscript for publication after English checking.
